# MXene Based Nanocomposites for Recent Solar Energy Technologies

**DOI:** 10.3390/nano12203666

**Published:** 2022-10-18

**Authors:** T. F. Alhamada, M. A. Azmah Hanim, D. W. Jung, R. Saidur, A. Nuraini, W. Z. Wan Hasan

**Affiliations:** 1Department of Scientific Affairs, Northern Technical University, Mosul 41001, Iraq; 2Department of Mechanical and Manufacturing Engineering, Faculty of Engineering, Universiti Putra Malaysia, Serdang 43400, Selangor, Malaysia; 3Advance Engineering Materials and Composites Research Center (AEMC), Faculty of Engineering, Universiti Putra Malaysia, Serdang 43400, Selangor, Malaysia; 4Department of Mechanical Engineering, Jeju National University, 1 Ara 1-dong, Jeju 690-756, Korea; 5Centre for Nano-Materials and Energy Technology (RCNMET), School of Engineering and Technology, Sunway University, Petaling Jaya 47500, Selangor, Malaysia; 6Department of Electrical and Electronic Engineering, Faculty of Engineering, UPM, Serdang 43400, Selangor, Malaysia

**Keywords:** MXenes, nanocomposites, solar cells, power conversion efficiency

## Abstract

This article discusses the design and preparation of a modified MXene-based nanocomposite for increasing the power conversion efficiency and long-term stability of perovskite solar cells. The MXene family of materials among 2D nanomaterials has shown considerable promise in enhancing solar cell performance because of their remarkable surface-enhanced characteristics. Firstly, there are a variety of approaches to making MXene-reinforced composites, from solution mixing to powder metallurgy. In addition, their outstanding features, including high electrical conductivity, Young’s modulus, and distinctive shape, make them very advantageous for composite synthesis. In contrast, its excellent chemical stability, electronic conductivity, tunable band gaps, and ion intercalation make it a promising contender for various applications. Photovoltaic devices, which turn sunlight into electricity, are an exciting new area of research for sustainable power. Based on an analysis of recent articles, the hydro-thermal method has been widely used for synthesizing MXene-based nano-composites because of the easiness of fabrication and low cost. Finally, we identify new perspectives for adjusting the performance of MXene for various nanocomposites by controlling the composition of the two-dimensional transition metal MXene phase.

## 1. Introduction

Scientists have opened the floodgates to researching clean, renewable energy in the past few years. They think solar energy is the most abundant energy source that can meet society’s needs, which come from economic growth [1]. Solar energy can be used for everything from heating water to generating electricity. Solar energy can be harvested using photovoltaic (PV) technology. Reports on the world’s solar photovoltaic electricity supplies say that PV technologies will provide around 345 GW and 1081 GW by 2020 and 2030, respectively [2,3]. Southern European countries have found solar energy to be the most cost-effective option available. The analysis’s resolution was set at the regional level. Sustainable development advocates often cite the importance of integrated renewables like building-integrated photovoltaics [4]. A technical–economic study has been conducted by Ali and Alomar to evaluate the productivity of grid-connected photovoltaic (PV) solar systems. This study proved that investment in the technology of PV systems is quite favourable [5].

Research of nanomaterials is currently at the forefront because of its potential to solve issues crucial to human existence in areas like the environment, energy, medicine, fuel, etc., [6]. For nanomaterials, zero-dimensional, one-dimensional, and two-dimensional classifications exist. Combining nanoparticles with other bulk materials can also generate three-dimensional nanomaterials. Numerous studies have shown that 2D nanomaterials are very important. Two-dimensional nanomaterials can also be used in optoelectronic devices because of their capacity to modify their optical characteristics through their thickness [7,8].

In 2011, Gogotsi et al. investigated transition-metal nitrides, or carbides (MXenes), as star materials from MAX phases, which are layered compounds similar to graphite with monoatomic A-element layers sandwiched between metallic electrically conductive and stiff MX-blocks. The new compound was called MXenes because the A element had been removed from the MAX phase and its structure was similar to graphene [9].

Commercialization of perovskite solar cells requires high efficiency and excellent stability. According to the work of Bati et al., functionalizing MXene nanosheets with cesium ions and incorporating them into the perovskite layer can result in devices with improved efficiency and increased thermal stability. Passivating perovskite defects using additive, interface, and compositional engineering has been accomplished to date [10,11].

Guo et al. achieved a 12% increase in efficiency by adding MXene to the perovskite precursor [12]. Agresti et al. later demonstrated that MXene could boost PSC PCE by 26% and reduce hysteresis when used as a dopant or interlayer. According to these conclusions, MXenes have the potential to serve as dopants in PSCs [13,14].

In order to better understand the physics and chemistry behind them, MXene-based materials for solar cell applications are categorized into various roles, including as electrodes, additives in perovskite solar cells, electron/hole transport layers, and MXene-silicon based heterojunction solar cells [15,16]. When MXenes are used in perovskite solar cells, the ion transport speed increases and the crystalline size is extended, which benefits energy conversion [8]. The solar cell properties of functional nanomaterials, such as 2D materials like MXene, are far superior to those of traditional materials. Energy conversion efficiency has been measured at 26.5%, three times greater than carbon nanotube silicon photovoltaic cells. Photovoltaics research has focused on 2D layered nanomaterials because of their unique properties. The complex roles that MXenes play in solar cell designs have been the biggest obstacle to their widespread use [17].

MXenes are a viable option for use in a wide variety of settings due to their adaptability and appealing characteristics. Its outstanding features, including high electrical conductivity, Young’s modulus, and distinctive morphology, make it very advantageous for composite construction [15]. Photovoltaic systems that turn solar energy into electrical energy have a promising future as a clean energy source due to the material’s good electronic conductivity, ion intercalation, chemical stability, and tunable band gaps. For photovoltaic conversion, nanostructured devices have evolved from the original wafer-based devices. These devices take advantage of a distributed heterojunction to produce and transfer charges in spatially separated phases. In most photovoltaic cells, two materials are used to separate the electron-hole pairs and charge transfers that lead to photocurrent generation [18]. Work functions for MXenes range from as low as 2.14 electronvolts to as high as 45.65 electronvolts. It was found that the work function was highly dependent upon the functional groups present on the surface, with the presence of –O groups increasing the work function and the presence of –OH groups decreasing it. When the barrier height is changed to an appropriate range for semiconductors or metal contacts, photocurrent can be generated [19]. A MXene-Si Schottky junction cell was fabricated by maintaining Mxene and Si in contact with one another in a vertical van der Waals heterostructure. This cell had an open voltage of 0.34 V and a current density of 12.9 mA cm^2^ under 100 mW cm^−1^. Strain engineering semiconductors could achieve excellent photoelectronic applications. A MXene-blue phosphorene heterojunction, for example, exhibits a valence band minimum from MXene and a conduction band minimum from phosphorene under conditions of moderate strain. In contrast, as the strain is increased, the band alignment is flipped. This heterostructured junction material will significantly benefit the development of photoelectronic and novel photonic applications [20]. For more than 1900 h, Chen et al. produced a solar cell with a high incipient power conversion efficiency of around 9.01% and long-term stability [18]. This electron is then transferred to a carbon electrode via a perovskite film, where it accumulates and is discharged through holes in the perovskite film. As shown in Figure 1, the charge transport arrangement and mechanics are depicted [21,22].

In recent years, the efficiency of power conversion in perovskite solar cells (PSCs) has continued to rise, attracting researchers’ interest [23]. First, Nb_2_C MXenes were introduced as an external addition to the SnO_2_ electron transfer layer (ETL) developed by Y. Niu et al., which led to a noticeable growth of SnO_2_ grains, as shown in Figure 2. They obtained a maximum power conversion efficiency of 22.86%, and these target devices maintained 98% of their initial efficiencies after 40 days at 25 degrees Celsius and 40–60 percent humidity [24].

The main objective of this paper was to describe in detail the fundamental principles for creating each 2D transition metal MXene structure-based nanocomposite and its tunable properties depending on the composition of transition metals. We offer an alternative approach to obtaining efficient PSCs by providing an in-depth mechanistic understanding of MXene interface engineering. Figure 3 shows a schematic overview of the different MXene NCPs discussed in this review paper, such as MXene-polymer, MXene-metal oxide, and MXene-carbon.

This paper summarizes all the documented work on incorporating MXene-based nanocomposites into recent solar energy technologies to enhance solar power generation and work stability. The following section describes perovskite-based solar cells. Section 3: Fabrication of MXene-nanocomposite and enumerates the many functions performed by MXene in solar cells. After that, the role of MXene surface termination groups. Section 5: MXene-based nanocomposites and how they are classified. A conclusion and prospects are given in Section 6.

## 2. Perovskite-Based Solar Cells

Since PVSK solar cells have such good light-harvesting qualities, they have developed rapidly in recent years, and numerous milestones have been attained in this sector, such as a high PCE of up to 23.2%, stability for more than a thousand hours, and so on. However, in order to meet its potential PCE limits (30–33%), a number of complex difficulties, such as the higher crystal size and fewer grain boundaries, must be handled. Two-dimensional MXene (Ti_3_C_2_T_x_) was initially suggested as an additive in PVSK solar cells by Guo et al. in their paper [12,25,26,27,28]. Inserting a Ti_3_C_2_-MXene has an energy level that is higher than the carbon electrode, which lowers PVSK’s conduction and valence band, thereby decreasing the pace at which the photocurrent is transferred and accelerating the transfer of the hole, which is represented in Figure 4a [21]. By inserting a thin layer of Ti_3_C_2_-MXene, it is possible to passivate the PVSK flake surface and create a direct conducting channel between Ti_3_C_2_-MXene and CsPbBr_3_, which speeds up carrier transport to the carbon electrode. Recently, 2D Ruddlesden–Popper PVSK solar cells have been suggested as a way to improve the long-term stability of operation. Jin et al. [29] demonstrated perovskite solar cells with Ti_3_C_2_T_X_ MXene-doped PVSK flakes, which increased the device’s current density. In addition, a MXene-MAPbBr_3_ heterojunction is formed using the in situ solution growth method. As shown in Figure 4b,c, the MXene-MAPbBr_3_ heterojunction’s charge and energy transfer speeds are significantly facilitated, positively contributing to the performance enhancement [30,31].

## 3. Fabrication of MXene Composite

Composites reinforced with MXene can be made in various ways, from solution mixing to powder metallurgy. MXene composites can be prepared using various methods, some of which are described below.

### 3.1. Solution Mixing

Solution mixing techniques have produced most MXene-reinforced polymer nanocomposites due to the hydrophilic character of MXene nanosheets supplied by the functional groups [32,33]. As shown in Figure 5, MXene nanoparticles are often distributed in polar solvents such as water [34], *N*,*N*-dimethylformamide (DMF) [35], and dimethylsulfoxide (DMSO) [36]. Due to their mutual solubility, polymer components might potentially be dissolved in the same dispersant or a different one [35]. These solutions, which consist of the polymer and MXene, are combined and blended to produce a homogeneous slurry of MXene composites. It should be emphasized that the solubility of MXene in nonpolar polymers or those with weakly polar groups is still problematic; thus, a proper surface pretreatment is required to improve dispersibility [37,38]. Solution mixing is a straightforward procedure that takes advantage of the hydrophilicity of MXene nanoparticles, but serious limitations, such as the formation of an abnormal quantity of environmental waste, poor mechanical qualities associated with the resulting composites, and laborious evaporation of solvents, generally prohibit its application [39].

### 3.2. Hydrothermal Process

The hydrothermal technique, also known as solvent thermal or solvothermal, is an often-documented procedure for producing a variety of new substances, new materials, and new compounds [41,42,43,44,45], especially MXene ceramic nanocomposites because of its simplicity, low cost, and widespread use. As shown in Figure 6, the amount of restacking required by this approach is low, and the resulting distributional uniformity is adequate [46,47,48,49]. For instance, BiFeO_3_ (BFO)/Ti_3_C_2_ nanohybrid was produced by using a straightforward and cheap double solvent solvothermal process for the break-down of organic dye and colourless contaminants [47]. In another study, tetrabutyl titanate Ti(OBu)_4_ was used in a straightforward hydrothermal process at a low temperature to create a Ti_3_C_2_/TiO_2_ composite [50]. High oxidation or interdiffusion is unavoidable due to the method’s use of excessive temperatures, and achieving a uniform dispersion of particles is difficult in comparison to other techniques [51,52].

The significant rise in PCE value is primarily attributable to the synergistic effects of the hydrothermal method and the one-of-a-kind layered morphology of conductive MXene nanosheets and their cocatalysts with CoS nanoparticles. These two factors contribute to the catalytic activity of the material. According to the findings of Chen et al., MXene-based composite CE materials show a great deal of promise for high electro-catalytic activity in QDSCs. These materials generate an abundant number of catalytic active sites, have good permeability, and exhibit outstanding charge transfer and ion-diffusion performance [53].

**Figure 6 nanomaterials-12-03666-f006:**
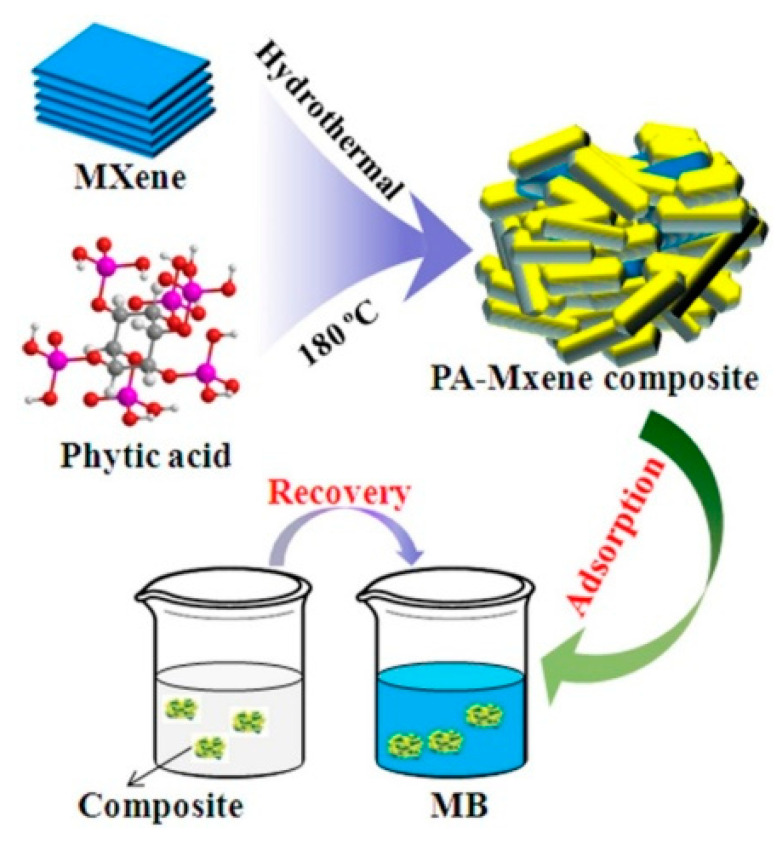
Schematic illustration of Mxene-based composites synthesized via the hydrothermal process. Reprinted with permission from Ref. [54]. 2020, Elsevier.

### 3.3. Powder Metallurgy

Powder metallurgy reduces waste, makes smooth surfaces, and the process produces less than 3% scrap. Tooling expenses, on the other hand, may be justified in large-scale manufacturing, as shown in Figure 7 [37,55] An aluminum (Al) matrix containing 10% Ti_3_C_2_T_x_ in a polypropylene container was tested for chemical stability using powder metallurgy [56]. After cold pressing, the pellet was sintered without applying pressure at a temperature between 500 and 7001 degrees Celsius [56]. Pressureless sintering followed by a hot extrusion technique was used by [57] to generate Ti_3_C_2_T_x_/Al with a MXene concentration of 0–3 wt%, while [37] used spark plasma sintering to produce Cu/Ti_3_C_2_T_x_ with improved tribological characteristics. Self-lubricating Ti_3_C_2_ nanosheet/copper (Ti_3_C_2_/Cu) composite coatings were studied by [58], who used an electrodeposition approach at room temperature to create the coatings using Ti_3_C_2_ nanosheets. Similarly, Refs. [58,59] developed a novel MXene-Ag nanowire composite using a simple electrodeposition approach [52].

## 4. Role of Surface Termination Groups

MXenes cleared the way for the possible construction of innovative optoelectronic devices based on developing surface termination groups. Surface termination groups may adjust the band gap without altering the Ti_2_CT_x_ MXene’s original structure, and this is a valuable technique for regulating the material’s electrical characteristics [60,61]. On the other hand, theoretical investigations have shown that surface termination groups affect the electronic structure of Ti_2_CO_2_ [62]. The pristine MXenes (Ti_3_C_2_) have a metallic structure. In contrast, Ti_3_C_2_(OH)_2_ terminated with –OH displays semiconducting properties [63]. As a result, surface functional groups (–OH and –F) show semiconducting behavior with a valence band–conduction band energy differential of 0.05–0.1 eV [20]. Enyashin et al. theorized that the band gap of –OH terminated Ti_3_C_2_ within the range of 0–0.042 eV [64]. While the work function of –O and –OH terminated Ti_3_C_2_ MXenes was shown by Schultz et al. [65]. According to the researchers, the kind of OH termination has little effect on the variations of strain energies in titanium carbide TiC_x_ nanotubes, but it does affect the relative stability of the planar parent phases, as shown in Figure 8 [64].

Controlling the amount of TiO_2_ and Ti_3_C_2_ in the resulting TiO_2_/Ti_3_C_2_ composite affects the separation of charge carriers based on: (i) the surface alkalization processes of pure Ti_3_C_2_; (ii) hydrothermal oxidation temperature; (iii) calcination temperature; (iv) surface termination groups (–F, –OH, or –O); and (v) hydrothermal reaction time [66]. The chemically reactive M-A bonding in the M_n+1_AX_n_ phase makes selective etching of the interleaved A element a viable option for separating M_n+1_AX_n_ layers. In 2011, Naguib et al. used a Ti_3_AlC_2_ MAX phase powder to investigate Ti_3_C_2_ MXenes (graphene-like morphology) [9].

## 5. MXene-Reinforced Nanocomposites

Combining MXenes with polymers, ceramics, metals, and nanoparticles yields composites with improved performance. Their exceptional optical, electrical, structural, mechanical, and thermal qualities result from their one-of-a-kind chemical and physical properties. Many other nanomaterials, including graphene derivatives, metal oxides, metals, and polymer monomers, have been successfully merged with MXene to create MXene-based hybrid nanocomposites, which improve upon the characteristics and practicality of pure MXene. Effective types for preparing MXene composites are as follows.

### 5.1. MXene-Metals/Ceramics Composite

MXenes are often employed to reinforce polymeric materials, but they can also be utilized to reinforce metallic or ceramic materials [67,68,69]. Reinforcement agents like graphene and CNTs have previously been tried in metals. On the other hand, metal matrix composites have faced significant difficulties due to agglomeration and poor wettability [70]. Pure MXene has been successfully combined with a wide range of nanomaterials, including graphene derivatives, metal oxides, and metals, to create Mxene-based hybrid nanocomposites [39].

### 5.2. MXene-Polymer Composite

Using MXenes, polymer-based composites get a significant advantage in mechanical performance [39,71,72]. MXenes offer a wide range of applications as composite components because of their unique chemistry [16,73,74]. MXenes could greatly affect how spherulites grow and how polymeric materials crystallize [75,76]. Since the MXene sheets have a high aspect ratio and the -OH termination groups provide hydrogen-bonding interactions, the Ti_3_C_2_T_x_ was found to significantly alter the glass transition temperature (Tg), and the mechanical strength increased by 23 percent, from 104.6 MPa for pure Nafion to 128.4 MPa for the composite sample [77]. Polymeric molecules respond better to MXene’s functional groups than to Graphene’s. These functional groups include the –O_2_, –OH, and –F. Graphene devoid of surface terminations is often insufficient for composite production [78]. Due to its hydrophilic nature, MXene sheets have excellent wettability with a broad range of materials. It makes it easy to disperse and spread the sheets in various liquids [79]. Currently, MXenes have been employed in several types of polymeric matrices, including polyurethane (PU) [71,80], polyacrylic acid (PAA) [81], polylactic acid (PLA) [16], poly-vinyl alcohol (PVA) [82], nylon-6 [83], chitosan [84], and polyvinylidene fluoride (PVDF) [35], etc.

## 6. Conclusions and Prospect

Technological development has come a long way since MXene’s discovery in 2011. In 2018, two-dimensional transition metal (MXene) contributed to the enhancement of solar cell manufacture by increasing the efficiency of produced energy and solar cells’ durability. MXene-based nanocomposites offer incredibly promising possibilities as a future for this remarkably fast-growing subject of nanotechnology and may be further investigated in many fields of science and technology. Substantial work is necessary to characterize and improve the characteristics of MXene-based nanocomposites, resulting in materials with superior desirable features. Even though many Mxene-polymer composites are synthesized, understanding how the microstructural characteristics of MXene-metal or ceramic composites influence their physical properties is still in its early stages. More research is required on how varying the concentration of a surface passivating functional group affects its properties. While several MXenes are readily accessible, Ti_3_C_2_Tx is now the most popular MXene used in solar cell fabrication. This paper reviews the recent progress in MXene-reinforced composites ranging from polymer-based materials to ceramic–or metal–matrix nanocomposites. We summarized the comprehensive studies on using MXene-based nanocomposites in solar cells and collated nearly all the results in the literature, as it has only been four years since MXene’s initial application in a solar cell was proven. More research is required on how varying the concentration of a surface passivating functional group affects its properties. The essential device parameters are listed in Appendix A [85,86,87,88,89,90,91,92,93,94,95,96,97,98,99,100]. 

In light of the above, the essential findings of this study are: MXenes interact with other materials to generate hybrids and nanocomposites with enhanced or extra properties. Applications for these novel materials in renewable energy, energy storage, and energy conversion are possible;As shown in Appendix A, MXenes, metal oxides, and noble metals have an impact on the device characterization data of solar cells. MXene has been used as a booster in nanocomposite for solar cells, which has dominated scientific research because conventionally designed PSCs are more efficient than metal oxide and noble metals;Incorporating 2D transition-metal MXenes into the category of 2D materials has improved the design choices for nanomaterials to meet the expanding technological demands;We have confirmed that the Nb_2_C MXenes were a suitable additive for the SnO_2_ ETL to make the PSCs work much better;Mechanisms that improve the performance of MXenes in solar cells are reviewed in depth for their future development and commercial use;Small Mxene (CoS) nanoparticles boost photovoltaic performance by generating excellent permeability, abundant catalytic active sites, ion-diffusion performance, and outstanding charge transfer.

## Figures and Tables

**Figure 1 nanomaterials-12-03666-f001:**
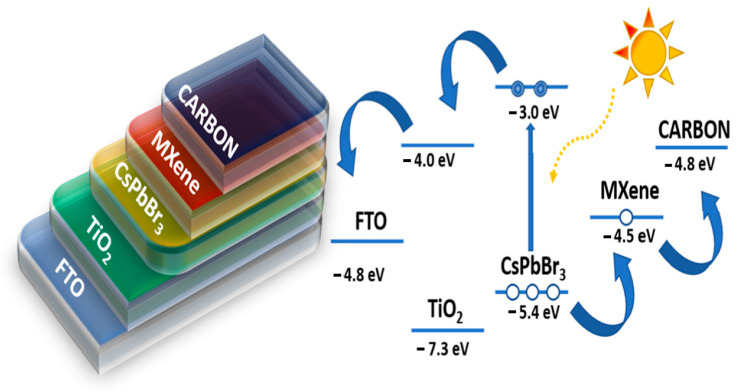
Shows the structure and charge transport mechanism of an FTO/TiO_2_/CsPbBr_3_/Ti_3_C_2_-MXene/carbon perovskite solar cell. Reprinted with permission from Ref. [22]. 2021, Elsevier.

**Figure 2 nanomaterials-12-03666-f002:**
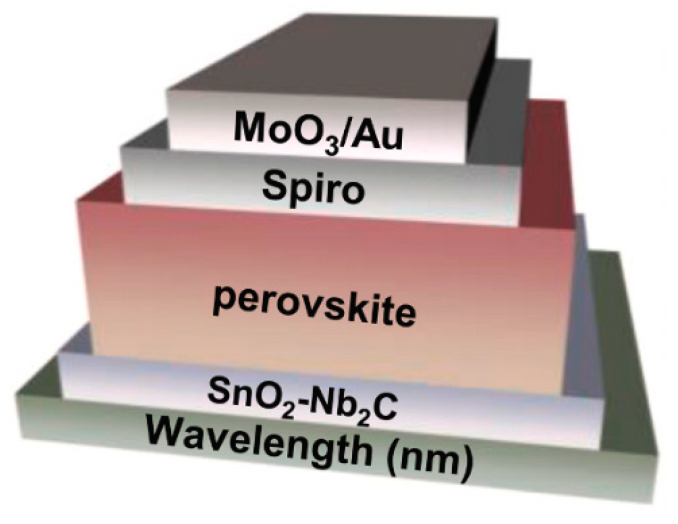
Schematic of the PSCs structure. Reprinted with permission from Ref. [24]. 2021, Elsevier.

**Figure 3 nanomaterials-12-03666-f003:**
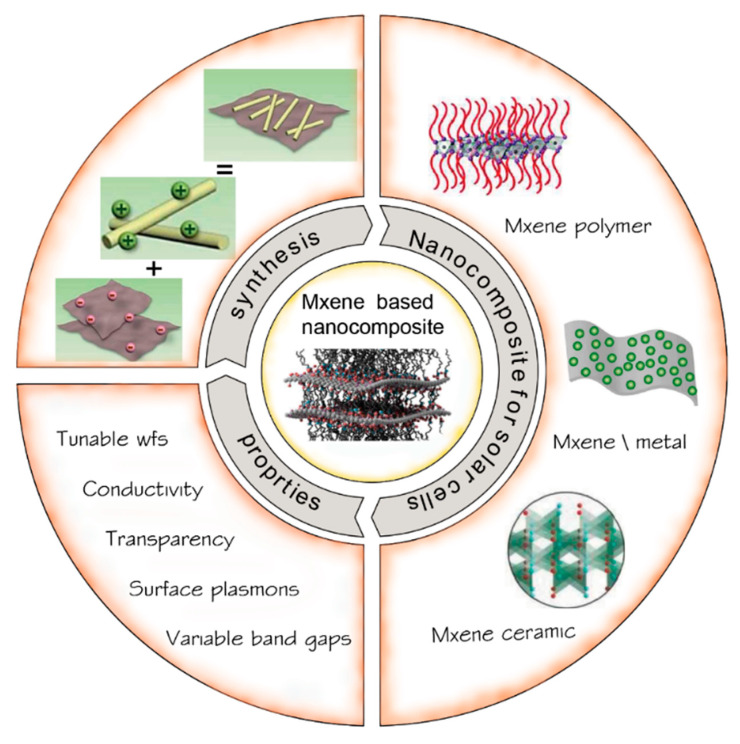
An illustration of a schematic overview of MXene-based nanocomposites for solar cell applications.

**Figure 4 nanomaterials-12-03666-f004:**
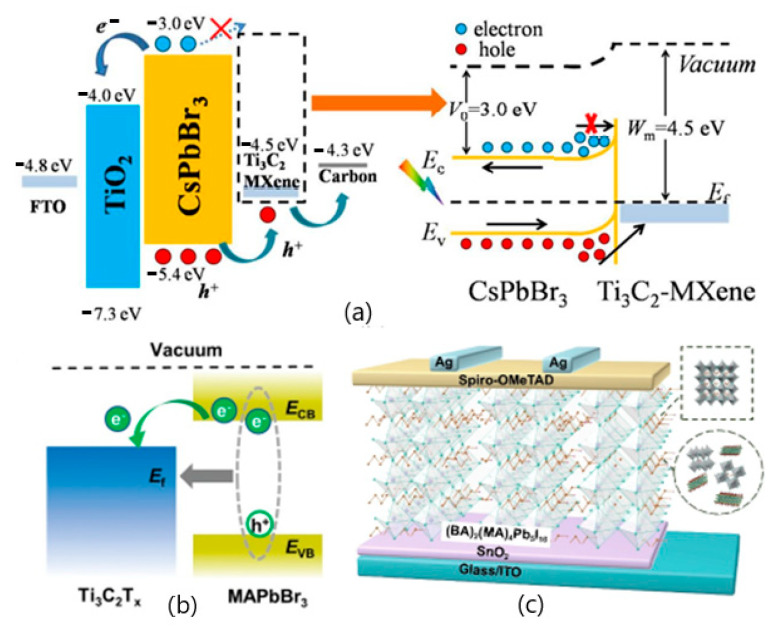
(**a**) The device’s energy bandgap and carrier transport mechanism under illumination at the interface, respectively. Reprinted with permission from Ref. [21]. 2019, Royal Society of Chemistry. (**b**) Energy transfer scheme between MXene and perovskite nanoflakes. Reprinted with permission from Ref. [29] 2021, Springer Nature. (**c**) The schematic diagram of the device. Reprinted with permission from Ref. [30]. 2020, John Wiley and Sons.

**Figure 5 nanomaterials-12-03666-f005:**
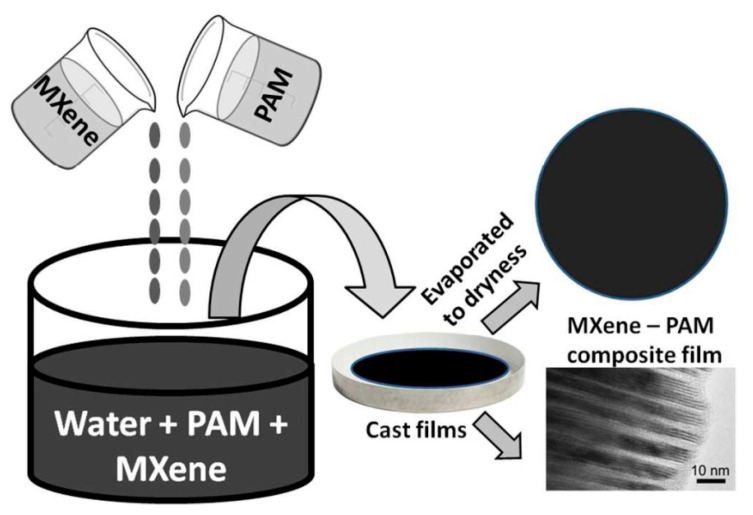
Schematic of the synthesis of MXene-Polyacrylamide nanocomposite films by the solution mixing method. Reprinted with permission from Ref. [40]. 2016, RSC Publishing.

**Figure 7 nanomaterials-12-03666-f007:**
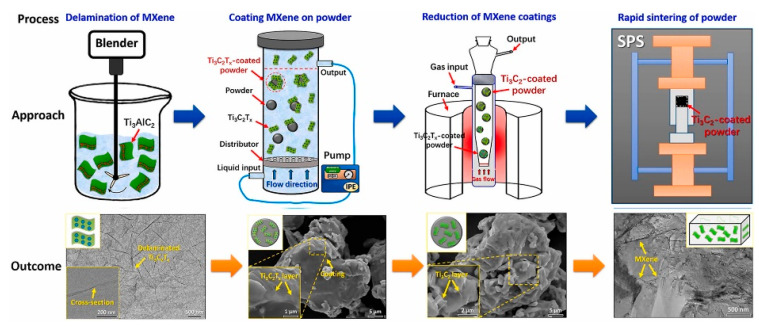
Schematic illustration of the powder metallurgy process. Reprinted with permission from Ref. [54]. 2020, Elsevier.

**Figure 8 nanomaterials-12-03666-f008:**
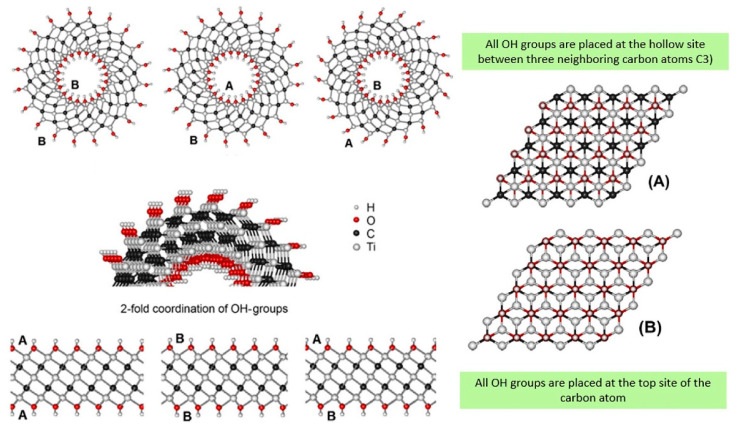
The arrangement of –OH termination at positions (**A**,**B**) in the inner and outer walls of Ti_3_C_2_T_x_ MXene nanotubes. Reprinted with permission from Ref. [64]. 2012, Elsevier.

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
