# Peer review of "MXene Based Nanocomposites for Recent Solar Energy Technologies"

_nanomaterials, 2022, doi:10.3390/nano12203666_

Round 1
Reviewer 1 Report
The manuscript presents review of investigation of the design and preparation of modified MXene-based nanocomposite for increasing the power conversion efficiency and long-term stability of perovskite solar cells. The fundamental principles of creation 2D transition metal MXenen structure-based nanocomposite and its tunable properties are described. The presented results are very interesting and practically useful.
The paper can be published.
Author Response
Thank you for the kind comments.
Reviewer 2 Report
Review for nanomaterials-1944409
This paper reviews the recent progress in MXene-reinforced composites for solar energy technologies, including introduction, perovskite-based solar cells, fabrication of MXene composite, role of surface termination groups, MXene-reinforced nanocomposites as well as conclusions and prospect. The core contents contain three aspects: (a) The efficiency of power conversion in perovskite solar cells; (b) Composites reinforced with MXene for solar cells and the fabrication methods; (c) Improve solar power generation and operational stability.
I think the main text, the supplementary information together with related schemes, figures and tables basically have been carefully prepared. Therefore, I recommend publication of the manuscript. However, some minor errors and mistakes should be considered and resolved in the final revision.
1. Line 22: correct “easy” as “easiness”;
2. Line 23: change “described as a method for ” into “used for”;
3. Line 41:modify “Research into” as “ Research of”;
4. Line 180-182: Solvo(hydro)thermal synthetic method has many excellent features, so it was often selected to prepare a variety of new substance, new materials and new compounds, including MXene ceramic nanocomposites. So please refine the sentence “The hydrothermal technique, also known as solvent thermal or solvothermal, is a often-documented procedure for producing MXene ceramic nanocomposites because of its simplicity, low cost, and widespread use.” As “The hydrothermal technique, also known as solvent thermal or solvothermal, is a often-documented procedure for producing a variety of new substance, new materials and new compounds [1-5], especially MXene ceramic nanocomposites because of its simplicity, low cost, and widespread use.” And the following references may be cited to support this opinion.
[1]J. Solid State Chem. 305 (2022) 122636. DOI: 10.1016/j.jssc.2021.122636.
[2] Inorg. Chim. Acta 530 (2022) 120697. DOI:10.1016/j.ica.2021.120697.
[3] Z. Naturforsch. B Chem. Sci. 2022, 77, 561-564. DOI: 10.1515/znb-2022-0031.
[4] Inorg. Chem. 2021, 60, 10109-10113. DOI:10.1021/acs.inorgchem.1c01541.
[5] New J. Chem. 41 (2017) 12611–12616. DOI:10.1039/C7NJ02021J.
5. Please give the correct subscript numbers for the molecular formula of the correspondig compounds, involving in Refs [9, 10, 18, 25, 28, 36, 45, 47-50, 56, 58, 66-68, 71, 77-80, 82]; Supplementary Materials, Table S2 and S3; refs.[2-8] and so on.
6. The couclusion section may be much longer. Please reprepare a new shorter and conise one.
Author Response
|
No. |
Comment |
Action |
|
1 |
Line 22: correct “easy” as “easiness”; |
The required correction in this comment has been done. |
|
2 |
Line 23: change “described as a method for ” into “used for”; |
The required correction in this comment has been done. |
|
3 |
Line 41:modify “Research into” as “ Research of”; |
The required correction in this comment has been done. |
|
4 |
Line 180-182: Solvo(hydro)thermal synthetic method has many excellent features, so it was often selected to prepare a variety of new substance, new materials and new compounds, including MXene ceramic nanocomposites. So please refine the sentence “The hydrothermal technique, also known as solvent thermal or solvothermal, is a often-documented procedure for producing MXene ceramic nanocomposites because of its simplicity, low cost, and widespread use.” As “The hydrothermal technique, also known as solvent thermal or solvothermal, is a often-documented procedure for producing a variety of new substance, new materials and new compounds [1-5], especially MXene ceramic nanocomposites because of its simplicity, low cost, and widespread use.” And the following references may be cited to support this opinion. [1]J. Solid State Chem. 305 (2022) 122636. DOI: 10.1016/j.jssc.2021.122636. [2] Inorg. Chim. Acta 530 (2022) 120697. DOI:10.1016/j.ica.2021.120697. [3] Z. Naturforsch. B Chem. Sci. 2022, 77, 561-564. DOI: 10.1515/znb-2022-0031. [4] Inorg. Chem. 2021, 60, 10109-10113. DOI:10.1021/acs.inorgchem.1c01541. [5] New J. Chem. 41 (2017) 12611–12616. DOI:10.1039/C7NJ02021J.
|
The required correction in this comment has been done. The references have been placed at the end of the paragraph, as lines#186-191; [42]-[46]. |
|
5 |
Please give the correct subscript numbers for the molecular formula of the correspondig compounds, involving in Refs [9, 10, 18, 25, 28, 36, 45, 47-50, 56, 58, 66-68, 71, 77-80, 82]; Supplementary Materials, Table S2 and S3; refs.[2-8] and so on. |
All the subscript numbers in the comment have been corrected. |
|
6 |
The conclusion section may be much longer. Please reprepare a new shorter and concise one. |
Thank you for your kind comments. The required correction in this comment has been done. as lines#293-309. |
Reviewer 3 Report
Please check English spell.
Please revise literature. Avoid lumping references but cite each source by its novel contribution or as a support for statements.
First references are too general. Please, if the authors want to introduce the general framework, include recent studies on solar energy such as statistics, applications, data, etc. Look at prof. Francesco Mancini 's work or prof. Andy van den Dobbelsteen 's ones.
There is no clear Results and Discussion section.
Author Response
|
No. |
Comment |
Action |
|
1 |
Please check English spell. |
The required correction in this comment has been done. |
|
2 |
Please revise literature. Avoid lumping references but cite each source by its novel contribution or as a support for statements. |
The required correction in this comment have been done. as lines#36, 39, 52, 62, 66. |
|
3 |
First references are too general. Please, if the authors want to introduce the general framework, include recent studies on solar energy such as statistics, applications, data, etc. Look at prof. Francesco Mancini 's work or prof. Andy van den Dobbelsteen 's ones. |
Thank you for the comment. We try to incorporate in our correction as line#40-45;[4]-[5]. |
|
4 |
There is no clear Results and Discussion section. |
We try to incorporate key items and discussion in each subtopic since this is a review paper. We try to be more concise in our conclusion as amended in lines#293-309. We hope that this is acceptable. |
Reviewer 4 Report
Thу paper review the design and preparation of modified MXene-based nanocomposites. This topic can be interesting for the community. Thus the paper can be accepted for publication after correcting the typos.
Author Response
|
Thank you for the kind comments. |
Round 2
Reviewer 3 Report
The author addressed all my concerns.